# Pharmacotherapy for Obstructive Sleep Apnoea and Coupled Life: A Descriptive Phenomenological Study of a Complex Interaction

**DOI:** 10.3390/healthcare10101859

**Published:** 2022-09-24

**Authors:** Debora Rosa, Elisa Perger, Giulia Villa, Loris Bonetti, Valeria Storti, Elisa Cartabia, Giuseppina Ledonne, Carolina Lombardi, Albanesi Beatrice, Carla Amigoni, Gianfranco Parati

**Affiliations:** 1Istituto Auxologico Italiano, IRCCS, Sleep Disorders Center & Department of Cardiovascular, Neural and Metabolic Sciences, 20149 Milan, Italy; 2Department of Medicine and Surgery, University of Milano-Bicocca, 20126 Milan, Italy; 3Center for Nursing Research and Innovation, Vita-Salute San Raffaele University, 20132 Milan, Italy; 4Nursing Research Competence Centre, Nursing Direction Department, Ente Ospedaliero Cantonale (EOC), 6500 Bellinzona, Switzerland; 5Department of Business Economics, Health and Social Care, University of Applied Sciences and Arts of Southern Switzerland, 6928 Manno, Switzerland; 6Department of Nursing, University of Milan, ASST Fatebenefratelli Sacco, Luigi Sacco Teaching Hospital, 20146 Milan, Italy; 7Neuromotor Rehabilitation Unit, Istituto Auxologico Italiano, IRCCS, 20149 Milan, Italy; 8Department of Clinical and Molecular Medicine, University “La Sapienza”, 00185 Rome, Italy; 9Istituto Auxologico Italiano, IRCCS, SITR Lombardia, 20149 Milan, Italy

**Keywords:** adherence, couple, nursing, OSA, pharmacotherapy, qualitative study, sleep

## Abstract

Background: Obstructive Sleep Apnoea (OSA) is a common chronic sleep-related breathing disorder. Drug therapy is a recent approach to treating OSA, and no data is available regarding its effects on a couple’s life. The aim of this study was to investigate the impact on couples’ lives of a novel drug treatment for OSA. Methods: Participants of a trial on new drug therapy (reboxetine plus oxybutynin) for OSA were interviewed. The study was conducted using a descriptive phenomenological approach by Sundler et al. Results: Ten patients and six of their partners were recruited through a purposive sampling between February and April 2021. The results indicate that drug treatment of OSA had a favourable impact on the couples’ lives. Pharmacotherapy improved self-reported sleep quality, as the absence of CPAP allows people to sleep closer to the bed partner and because the patient does not have to sleep in a forced position. Moreover, the couples developed teamwork from the beginning which appeared to be crucial during the trial, especially when some difficulties and adverse events occurred. Conclusions: This study presents the presence of a positive closed loop that could be considered a predictor of adherence to drug therapy.

## 1. Introduction

Obstructive Sleep Apnoea (OSA) is a common chronic sleep-related breathing disorder, characterised by episodes of complete (apnoea) or partial (hypopnoea) obstruction of the upper airway during sleep [1]. It is prevalent in the general population at between 40 and 85 years of age in men (49.7%) compared to women (23.4%) [1].

Besides work and traffic accidents due to daytime sleepiness, OSA is also related to higher comorbidities and cardiovascular mortality [2,3,4,5]. The presence of OSA correlates with increased endothelial dysfunction, platelet aggregation, coronary artery disease, and arterial stiffness. In addition, sleep fragmentation and intermittent hypoxia are associated with changes in lipids and increases in sympathetic activity, blood pressure, and plasma low-density lipoprotein (LDL) levels [6,7]. There are several therapeutic approaches to treating OSA, such as behaviour modification [8], weight loss [9], oral appliance therapy, surgical procedures [9], and use of continuous positive airway pressure (CPAP) [10].

To date, CPAP is the gold standard in the treatment of OSA [11,12]. However, positive airway pressure’s tolerance is often difficult and adherence varies from 46% to 83% [8,9,13,14,15]. Alternatives to CPAP are being investigated and one of the most promising is drug therapy [16,17,18,19]. A double-blind randomised controlled crossover trial showed for the first time that pharmacological resolution of OSA is possible using a combination of noradrenergic (atomoxetine) and antimuscarinic (oxybutynin) drugs that have an effect on the dilating muscles of the upper airways during sleep [19]. Based on these results, another double-blind randomised controlled crossover trial with placebo was conducted in Italy to evaluate the efficacy of a combination of reboxetine and oxybutynin (reb-oxy) on the severity of OSA over 1 week of treatment (RebOx) [17]. The results confirm and highlight potential pharmacotherapy possibilities for the treatment of OSA.

OSA is a problem that also affects partners [20,21]. Many patients sleep with a stable partner, and OSA often is characterized by loud snoring followed by periods of silence when breathing stops or nearly stops. Eventually, this reduction or pause in breathing may be followed by an arousal, and the subject might awaken with a loud snort or gasping sound [22]. Examining the perspectives and experiences of patients and bed partners regarding OSA and the related therapy is essential to gaining a better understanding of the factors that facilitate treatment adherence. Some studies investigated variables associated with CPAP treatment from the patients’ point of view and, importantly, from the point of view of their partners [23,24,25], but no one has yet investigated drug therapy as it is still experimental. To our knowledge, this is the first qualitative study involving both patients and their partners to better understand the experience of drug therapy and its influence on a couple’s life. The aim of this study was to investigate the impact on couple life of a novel drug treatment for OSA.

## 2. Materials and Methods

### 2.1. Study Design

The study was conducted using a descriptive phenomenological approach by Van Manen [26], and the authors analysed data following qualitative thematic analysis by Sundler et al. [27]. According to this approach, the researcher must be open to the meaning of the interviewee’s lived experiences. In fact, the researcher must assume a position as an attentive observer and be sensitive to the communication of experiences and be able to question the understanding of the data, through a self-reflective attitude that allows him/her to acknowledge his/her pre-understanding. The authors suggest replacing bracketing with the construction on the question of a representative way to describe the meaning of something [27].

Participants to the RCT RebOx [17] were males or females between 18 and 70 years of age and a diagnosis of moderate-to-severe OSA with CPAP intolerance or poor compliance (compliance was defined as use of CPAP 4 h per night for 70% of nights) or CPAP-naïve. RebOx was a Phase II, randomised, double-blind, placebo-controlled, cross-over, single-centre efficacy study of the Reb-Oxy combination in adults with OSA. Eligible participants were randomised to first receive 4 mg reboxetine plus 5 mg oxybutynin or a matching placebo. Subjects took the study drug at home for seven days. A washout period of 7–10 days was followed by a switch to the other arm of the study. The study was conducted as a quadruple-blind study (subjects, healthcare professionals, investigators and outcome assessors were blinded to treatment allocation) [17].

Patients were included in the present study and interviewed if they had knowledge and mastery of the Italian language, were able to understand the nature of the study. Partners were living with a participant to the clinical trial RebOx. The exclusion criteria for patients and partners was the presence of clinically relevant cognitive dysfunction.

### 2.2. Sampling

Participants were recruited at the Istituto Auxologico Italiano, in northern Italy, from the list of participants of the RebOx study and were recruited through a purposive sampling [27]. The data collection was carried out through semi-structured in-depth interviews using video-conferencing software or face-to-face interviews if possible. Those who agreed to participate were contacted to arrange a videoconference or a personal interview, depending on their availability [28]. During the interview, the research team member took notes and data regarding the interviewee’s non-verbal communication, such as motor, psycho-biological, and verbalisation aspects with field notes.

### 2.3. Structure of the Interviews

The interview consisted of open-ended questions with the aim of generating rich and detailed narrative responses from the participants. The interviews were conducted by a female researcher with a PhD in Nursing and Public Health and experience in qualitative research and working as a nursing researcher. During the interviews, another researcher participated as an observer in order to collect field notes. Each participant was interviewed individually, after obtaining informed consent, using video-conferencing platforms (Skype, Teams, Zoom) [28].

### 2.4. Data Analysis

Data analysis was carried out in three stages: to achieve familiarity with the data through open-minded reading, to search for meanings and themes, and organizing themes into a meaningful wholeness. In the first phase, the researchers were required to read the text to become familiar with the data, to explore experiences, to search for unique and novel sides rather than what is already known. The search for meanings must be carried out searching for the meanings of experiences, marking meanings, describing meanings with a few words and notes in the margins, comparing differences and similarities between meanings, organizing meanings in patterns, and from patterns, themes begin to emerge. Whereas the third phase the findings are written and rewritten while organizing meanings, themes are described in a meaningful text, and the explicit naming of the themes must be described in the experiences of the actual context [27].

Interviews were analysed by manual coding. The accuracy and rigor of the data was ensured by the cross-checking of two researchers: the first researcher transcribed the data collected; the second researcher listened to the audio recording while reading the transcript to ensure that it accurately reflected the words of the interviewee. Any information that could make the interviewee recognizable was edited to protect their identity. For this reason, pseudonyms were used instead of names of people, cities, streets, and organizations.

### 2.5. Ethical Considerations

The study was approved by the Ethics Committee and was developed in accordance with the Helsinki Declaration and the national ethical principles for scientific research and obtained the approval of the centres involved. All participants were informed of the purpose and methodology of the study by signing an informed consent form.

## 3. Results

Interviews with an average duration of 20 min were conducted between February and April 2021. The characteristics of the participants are described in Table 1 and Table 2. All 18 patients enrolled in the RebOx study, including the 2 that dropped out, and their partners were invited to participate. Eight patients refused to participate so that ten patients were interviewed (six face to face, and four in videoconference). None of the dropouts agreed to participate. Of those who were included, six partners agreed to be interviewed (two face to face, and four in videoconference). The following two themes were identified, “drug therapy helps the couple’s relationship” and “motivation for drug therapy outweighs side effects”. These themes were identified by both patients and partners (Table 3).

### 3.1. Drug Therapy Helps the Couple’s Relationship

This theme was identified by both patients and partners. Patients said that drug therapy improved their sleep quality. This also positively affected their partner’s sleep quality. This theme consisted of two subthemes: drug therapy and partner quality of life; support during drug therapy.

#### 3.1.1. Drug Therapy and the Couple’s Quality of Life

Patients report that drug therapy improved the quality of life as a couple as there was less snoring and they felt less nervous during the day. As a result, the partner was also more relieved not to hear the snoring and to know that they could finally sleep together in the same bed.

A patient who has never used CPAP said: “Not having used the CPAP device before, my quality of life seemed positive! And I did not feel nervous. I see my partner more relieved now [during the trial], at least she tells me at this moment. She does not hear me snoring anymore or going maybe on the sofa… absurdly, I felt asleep in the evening (on the sofa?) before going to bed and now she doesn’t live this thing anymore” (Patient OSA therapy naive).

Partners also reported that they benefited during the period of drug therapy. In fact, some found their sleep more peaceful, being able to sleep in the same bed without the noise and clutter of CPAP.

“I was more serene, of course […] I was more serene to the rest of her sleep”.(Partner)

“The serenity of sleeping and also feeling a little bit more relaxed in his sleep”.(Partner)

“Yes, I was glad that he maybe didn’t use the CPAP, so he was moving better in bed. And obviously he is happier without CPAP”.(Partner)

#### 3.1.2. Support during Drug Therapy

Most patients reported that their partners were supportive during the trial. Partners provided encouragement and sometimes suggested that the patient contact the trial doctor to participate in the trial, highlighting an important teamwork within the couple.

“[My partner] always agreed [to participate in the drug study]”.(Patient not-adherent)

“I was motivated, I was July and it was hot… it was very stressful for him. So, no, I kept telling him ‘come on, you will see we will have a result’. I motivated him a lot”.(Partner)

“I was the one who pushed him to go all the way. In the sense that the first visit he did to diagnose the OSA, that they put that monitoring [polysomnography] to keep in the house to see that the levels were right? I told him look you have to do something because apnoea is dangerous!”(Partner)

### 3.2. Motivation for Drug Therapy Outweighs Side Effects

This theme was identified in the interviews of both patients and partners. Three sub-themes were identified: individual responses to drug therapy, positive reaction to drug therapy, and sharing the therapy pathway.

#### 3.2.1. Individual Responses to Drug Therapy

This subcategory would appear to highlight the main perceived side effects of drug therapy for OSA. However, the positive side of these side effects was that they all consisted of transient situations, i.e., they were short-lived and did not persist beyond the morning.

“Yes! Because the effects, if they are not, let’s say, permanent and always present, I think that over the time that one has it, he no longer even pays attention to it. It is important that you sleep well”.(Patient 1, not-adherent)

“Yes, I would definitely recommend it. I would take it too, because the side effects were almost zero for me, so definitely I take it!”(Patient OSA therapy naïve)

“I hardly ever had any effects, only one was the smell of the urine it was a bit strange”.(Patient OSA therapy naïve)

For the partners, the patient’s health is more important than the adverse events that occurred during the drug trial.

“He says that maybe these tablets also affect sex […] On the other hand, I told him “You don’t have to worry about this, the important thing is that you recover and you are well”.(Partner)

#### 3.2.2. The Positive Reaction to Drug Therapy

Both patients and partners, in spite of the perceived transient side effects, stated that if drug therapy were on the market, they would recommend it to other people suffering from sleep apnoea, in order to help them and to look for an alternative solution to CPAP that could solve this problem.

“The first week I didn’t feel much effects, but over time was a little better [they both slept better and more] […] I felt quieter anyway”.(Partner)

## 4. Discussion

The aim of this study was to understand the experience of drug therapy and its influence on a couple’s life.

OSA can compromise a couple’s life due to daily nervousness about not sleeping in the same room because of loud snoring [20,22]. OSA negatively affects the partner, creating frustration, exhaustion, and interference at work [20]. For both members of the couple, the night time and daytime consequences of OSA and the decrease in intimacy due to not sharing a bed put a strain on the relationship [22]. This seems to be resolved by the use of drug therapy, which allows both to improve sleep quality. In particular, patients reported that they noticed a reduction in irritability in their partners due to sleep loss because it was possible for them to resume sharing a bed and have a better marital life quality during treatment. Both members of the couple expressed that during the pharmacological therapy they noted improvements in sleep, mood, and reduced daytime sleepiness. One aspect not to be overlooked, highlighted by many of the couples interviewed, is that drug therapy allows them to regain physical closeness during the night, because sleeping without the presence of the CPAP or loud snoring is liberating for the couple.

Some participants prior to the study were using CPAP as a therapy for OSA, with poor adherence or tolerance. Among the main causes of intolerance there are difficulties associated with the adjustment and comfort of the mask, and noise for bed partners [10,23,25]. For the majority of the interviewed, the experimental drug therapy with reboxetine and oxybutynin used in the trial was a viable alternative to CPAP. Nine participants stated that they would prefer pharmacological treatment than CPAP, as drugs are less invasive, allowing patients to sleep in any position and partners not to hear any noise. Among the four who were using CPAP before the trial, three wanted to proceed with drug therapy than going back to the CPAP treatment.

The bed partner was found to be supportive even before the beginning of the drug treatment. Teamwork developed in the couple from the very beginning and proved to be crucial during the trial, especially when some difficulties and adverse events occurred [22,29,30,31]. In spite of some temporary side effects, such as urination disorders, impotence, and dry mouth [17], the patients acknowledged that they had benefits from the drug treatment: better sleep, more energy, and high motivation to continue this pharmacotherapy. This qualitative study highlighted the necessity of the active participation of the bed partner in the chronicity of a pathology such as the one under study. The bed partner assumes a fundamental role for the subject in the adherence to therapy for example by supporting the patients when adverse events occurred.

The importance of the bed partner for the patient is also fundamental as the support and inclusion of the spouse in the chronicity of a pathology such as OSA plays a key role within the couple [21]. A positive closed loop interaction develops within the couple. Thus, teamwork encourages patients to participate in the study and to be adherent to the drug therapy. Thanks to the improvement in quality of sleep prompted by drug therapy, there is also an amelioration of the couple’s quality of life, as the treatment allows both partner and patient to enjoy time together and closeness at night. One of the consequences of the positive closed-loop response during the RebOx study is the positive, empathetic, and caring attitude of the partners [32] that sustained the patients’ motivation to continue drug therapy both in the presence of side effects and in moments of discouragement during the four-week duration of the study.

Future in-depth evaluations of the phenomenon of mutuality throw objective assessment scales are needed, especially in patients with OSA [33,34,35,36,37]. The investigation of possible strategies to strengthen the couple to promote adherence also need to be addressed in future clinical research [15,22,38,39].

## 5. Conclusions

The results of this qualitative study would seem to indicate that the couples’ lives were positively influenced during the use of reboxetine plus oxybutynin for the treatment of OSA in the RCT RebOx. In addition, bed partners saw improvements in the quality of sleep of the patients with a significant reduction of apnoea, less snoring, more serenity, and more alertness.

The preliminary results of this study and of Perger et al. [17] provide researchers with useful data for the design of quantitative studies with a larger sample size and prolonged duration. Furthermore, it is important that physicians, nurses, and technicians carefully assess both the patient and the partner, as the dyad, and not the individual, could be the integral component in improving and maintaining therapy adherence [40,41].

## Figures and Tables

**Table 1 healthcare-10-01859-t001:** Social-demographic characteristics.

	Patient (*n* = 10)	Partner (*n* = 6)
**Gender**		
Female	1	5
Male	9	1
**Age (mean, range)**	54.8 (43–62)	58 (44–68)
**Country**		
Italy	10	6
Other	0	1
**Education**		
Secondary school	3	4
High school	7	1
Bachelor’s degree		1
**Marital status**		
Married	7	4
Living together	2	1
Civil union	1	1

**Table 2 healthcare-10-01859-t002:** Patient/partner characteristics.

Relationship	Gender	Age	OSA Treatment
Patient	Female	61	Not-adherent to CPAP
Patient	Male	62	OSA therapy naïve
Patient	Male	55	Not-adherent to CPAP
Patient	Male	55	Not-adherent to CPAP
Patient	Male	54	OSA therapy naïve
Patient	Male	62	Not-adherent to CPAP
Patient	Male	60	OSA therapy naïve
Patient	Male	48	OSA therapy naïve
Patient	Male	46	OSA therapy naïve
Patient	Male	43	OSA therapy naïve
Partner	Female	56	Unknow
Partner	Male	68	Unknow
Partner	Female	60	Unknow
Partner	Female	58	Unknow
Partner	Female	44	Unknow
Partner	Female	55	Unknow

**Table 3 healthcare-10-01859-t003:** Themes, subthemes, and quotations.

Theme	Sub-Themes	Patient’s Quotations	Partner’s Quotations
1. Drug therapy helps the couple’s relationship	*1.1 Drug therapy and the couple’s quality of life*	“The use of drug therapy has also helped my partner sleep well and as a result has positively affected tour life as a couple”. (Patient 1, not-adherent)	“Yes, relatively in the sense that… the first night obviously there was little sleep, however the fact that it was in the same bed…” (Partner)
	*1.2 Support during drug therapy*	“We had talked about it and since I agreed to do it, she did too. Jointly, we came to the decision to participate in the study in essence”. (Patient OSA therapy naïve)	“If for your own, good I said ‘do it because at least you get a good night’s sleep, first of all, and then let me get a good night’s sleep too because if not I’m really going to go off the deep end with you too”. (Partner)
2. Motivation for drug therapy outweighs side effects	*2.1 Responses to drug therapy*	“Always positive experience, except however one day, when I felt a sensation just of anxiety, then fortunately I didn’t feel any other strange sensations, and anyway at the level of sleep I always felt refreshed”. (Patient OSA therapy naïve)	“The first tablets he took results were not seen, because nothing had changed. So, we had anyway imagined that “if this is the therapy, it does not work”, but since there was also the placebo we said, “let’s go ahead and see how it turns out”. In fact, then with the second pill the results were seen, in short”. (Partner)
	*2.2 The positive reaction to drug therapy*	“Yes, I would definitely recommend it. I would take it too, I mean in the sense if I have to choose, and logically the side effects are the ones that I’ve tried that have basically been almost zero for me, definitely!” (Patient OSA therapy naïve)	“I have to say that [during the period of drug therapy] my husband has been pretty easy going”. (Partner)

## Data Availability

Not applicable.

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
