# Peer review of "Pharmacotherapy for Obstructive Sleep Apnoea and Coupled Life: A Descriptive Phenomenological Study of a Complex Interaction"

_healthcare, 2022, doi:10.3390/healthcare10101859_

Round 1

Reviewer 1 Report (Previous Reviewer 1)

The authors aim to investigate the impact on the couple's life based on the usage of novel drug treatment for OSA. A new drug therapy (rebox- 31 etine plus oxybutynin) was prescribed. The authors found that ten patients and six of their partners 33 were recruited of which the drug treatment has a favourable impact on the couple's life.   

The study is interesting. However, the authors need to address a few concerns as listed:

1. Justification of the usage of 32 phenomenological approach by Sundler et al.

2. Justify the long-term approach of the novel drug, and how it is being studied

3. The measure used to assess the couples life/quality of life of individual

4. To have a control group with Cpap and other OSA treatments

5. Correlation between OSA grading and the overall well-being

Author Response

The authors aim to investigate the impact on the couple's life based on the usage of novel drug treatment for OSA. A new drug therapy (rebox- 31 etine plus oxybutynin) was prescribed. The authors found that ten patients and six of their partners 33 were recruited of which the drug treatment has a favourable impact on the couple's life.   

The study is interesting. However, the authors need to address a few concerns as listed:

  1. Justification of the usage of 32 phenomenological approach by Sundler et al.

Thank you for you comment. The authors added the justification in “Material and Methods” (line 83-91)

  1. Justify the long-term approach of the novel drug, and how it is being studied.

Thank you for your comment. In the conclusion authors added the long-term approach of the novel drug and how it is being studied (Line 325-326)

  1. To have a control group with Cpap and other OSA treatments

Thank you for your comment.

RebOx was a cross-over, randomized, double-blind, placebo-controlled, Phase II, single-center efficacy study of the Reb-Oxy combination in adults with OSA (as the authors mentioned in the introduction) (EudraCT Number: 2019-004917-15, Perger et al, 2022). The patients were therefore their own control and this did not allow the researchers to interview the people in the control group. In materials and methods, the authors have described the study design (line 95-101).

  1. The measure used to assess the couples life/quality of life of individual + 5. Correlation between OSA grading and the overall well-being

The qualitative research method studies phenomena in their natural contexts, attempting to make sense of them or interpret them in terms of the meaning the participant gives to them. It is used to investigate experiences, behaviour, usually collected through interviews with open-ended and non-predetermined response questions. This is why the data collected in a qualitative analysis cannot be measured directly, but must be interpreted (Morse, J.M. (1994). Designing funded qualitative research. In N.K. Denzin, & Y. S. Lincoln (Eds.), Handbook of Qualitative Research (pp. 220-225). Thousand Oaks, CA: SAGE; Morse, J. M., Barrett, M., Mayan, M., Olson, K., & Spiers, J. (2002). Verification strategies for establishing reliability and validity in qualitative research. International journal of qualitative methods, 1(2), 13-22; Sasso, L., Bagnasco, A., Ghirotto, L. (2015). La ricerca qualitativa. Una risorsa per i professionisti della salute. Milano: Edra).

Reviewer 2 Report (New Reviewer)

In this manuscript  there  is just changing point of view: in previously published article on novel pharmacological method of treatment the scientific data are presented and in this manuscript the same problem is considered from the patients’ and their families’ point of view. Thus, it is interesting because it describes impressions after introducing novel drugs into treatment of OSA patients. And it is not relevant as it carries minimal scientific impact.

Presented for the review study is a continuation of recently published paper in Chest journal, where the Authors have described novel pharmacological treatment of sleep apnea patients. So no – it is not original, as now this is kind of changing the perspective – from scientific point of view (as published earlier) to patient’s and their families’ point of view – as in this manuscript.  It does not add much knowledge. It describes impressions of the patients and their families related to novel treatment. So again: scientific soundness of this paper is minimal.

The text is very well written, clear, easy to read. But the material of the study consists of very few persons.

In summary: The paper is written by experienced Authors, there are no methodological errors, it describes in narrative way (even with citations of the subjects opinions) the impressions after taking novel drugs only by several patients the main results of this novel method of treatment have been published previously.  Although it might be interesting to read if the novel drug will be approved and available. There is no scientific soundness in this paper. One small notice: were there any face-to-face interviews (as stated in v.95) or each patient was interviewed using videoconference... (as stated in the v. 105/106)?

Author Response

  • In this manuscript there is just changing point of view: in previously published article on novel pharmacological method of treatment the scientific data are presented and, in this manuscript, the same problem is considered from the patients’ and their families’ point of view. Thus, it is interesting because it describes impressions after introducing novel drugs into treatment of OSA patients. And it is not relevant as it carries minimal scientific impact.
    Presented for the review study is a continuation of recently published paper in Chest journal, where the Authors have described novel pharmacological treatment of sleep apnea patients. So no – it is not original, as now this is kind of changing the perspective – from scientific point of view (as published earlier) to patient’s and their families’ point of view – as in this manuscript.  It does not add much knowledge. It describes impressions of the patients and their families related to novel treatment. So again: scientific soundness of this paper is minimal.
    The text is very well written, clear, easy to read. But the material of the study consists of very few persons.
    In summary: The paper is written by experienced Authors, there are no methodological errors, it describes in narrative way (even with citations of the subjects opinions) the impressions after taking novel drugs only by several patients the main results of this novel method of treatment have been published previously.  Although it might be interesting to read if the novel drug will be approved and available. There is no scientific soundness in this paper. One small notice: were there any face-to-face interviews (as stated in v.95) or each patient was interviewed using videoconference... (as stated in the v. 105/106)?

Thank you for your comment.

The qualitative research method studies phenomena in their natural contexts, attempting to make sense of them or interpret them in terms of the meaning the participant gives to them. It is used to investigate experiences, behaviour, usually collected through interviews with open-ended and non-predetermined response questions. This is why the data collected in a qualitative analysis cannot be measured directly, but must be interpreted (Morse, J.M. (1994). Designing funded qualitative research. In N.K. Denzin, & Y. S. Lincoln (Eds.), Handbook of Qualitative Research (pp. 220-225). Thousand Oaks, CA: SAGE; Morse, J. M., Barrett, M., Mayan, M., Olson, K., & Spiers, J. (2002). Verification strategies for establishing reliability and validity in qualitative research. International journal of qualitative methods, 1(2), 13-22; Sasso, L., Bagnasco, A., Ghirotto, L. (2015). La ricerca qualitativa. Una risorsa per i professionisti della salute. Milano: Edra).

In qualitative design, sampling is non-probabilistic and convenience sampling, chosen on the basis of the desirability and ease of accessing the field and recruiting participants who meet the inclusion criteria (Sandelowsky, M. (1995). Sample Size in qualitative research. Research in nursing and health, 18 (2), 179-183).

This is the first paper about drug therapy and couple’ life, and The authors think that this paper could lead to important new perspective on this treatment and how it can influence the life of a couple.

We added the number of patients and partners interviewed face-to-face or using video-conferencing software (lines 140-142)

Reviewer 3 Report (New Reviewer)

Minor corrections to improve the overall quality:

- among several comorbities OSA correlate strongly with obesity, gerd and ASthma, please discuss and cite doi:10.1016/j.jaip.2021.09.003.

- line 60, although cpap represents to date the initial treatment for OSA several surgical treatment are available for younger patients with palate or tongue base collapse. please discuss and cite doi: 10.3390/bs11120180.

-  line 63, the barbed surgery demonstrated interesting results both in subjective questionnaire as ESS than respiratory index, even after extrusion of the barbed sutures. please discuss and cite doi:10.1002/lary.29357.  and doi:10.1016/j.amjoto.2021.102994.

Methods Results

add a flow diagram for the study protocol

add p value as possible always in the tables.

Author Response

Minor corrections to improve the overall quality:

  • Among several comorbities OSA correlate strongly with obesity, gerd and ASthma, please discuss and cite doi:10.1016/j.jaip.2021.09.003.
  • Line 60, although cpap represents to date the initial treatment for OSA several surgical treatment are available for younger patients with palate or tongue base collapse. please discuss and cite doi: 10.3390/bs11120180.
  • Line 63, the barbed surgery demonstrated interesting results both in subjective questionnaire as ESS than respiratory index, even after extrusion of the barbed sutures. please discuss and cite doi:10.1002/lary.29357.  and doi:10.1016/j.amjoto.2021.102994.

Thank you for your comments. The authors have added new information (lines 50-72)

Methods Results

  • Add a flow diagram for the study protocol

Thank you for your comment. The authors added the flow diagram (Figure 1)

  • Add p value as possible always in the tables.

The qualitative research method studies phenomena in their natural contexts, attempting to make sense of them or interpret them in terms of the meaning the participant gives to them. It is used to investigate experiences, behaviour, usually collected through interviews with open-ended and non-predetermined response questions. This is why the data collected in a qualitative analysis cannot be measured directly, but must be interpreted (Morse, J.M. (1994). Designing funded qualitative research. In N.K. Denzin, & Y. S. Lincoln (Eds.), Handbook of Qualitative Research (pp. 220-225). Thousand Oaks, CA: SAGE; Morse, J. M., Barrett, M., Mayan, M., Olson, K., & Spiers, J. (2002). Verification strategies for establishing reliability and validity in qualitative research. International journal of qualitative methods, 1(2), 13-22; Sasso, L., Bagnasco, A., Ghirotto, L. (2015). La ricerca qualitativa. Una risorsa per i professionisti della salute. Milano: Edra).

Round 2

Reviewer 1 Report (Previous Reviewer 1)

The authors have revised the manuscript. However, there are still a few concerns regarding this study.

1. How do authors justify the efficacy of novel drug therapy for OSA without an objective test, as it is not ethically sound?

2. The authors state that: The results indicate that drug treatment of OSA has a favourable impact on couple's life. This study only has a very small sample size. To claim such statement appears rather dangerous.

Author Response

  1. How do authors justify the efficacy of novel drug therapy for OSA without an objective test, as it is not ethically sound?

Thank you for your new comment. The authors changed the conclusion (line 294-300).

  1. The authors state that: The results indicate that drug treatment of OSA has a favourable impact on couple's life. This study only has a very small sample size. To claim such statement appears rather dangerous.

Thank you for your new comment.

The sample size in qualitative research does not only refer to the number of people, but to the number of events sampled so it is the events that are narrated that constitute the sample. People are certainly central but enter qualitative studies primarily by virtue of their direct and personal knowledge of some event, which they are able and willing to communicate to others, and only secondarily by virtue of demographic characteristics. Sampling is purposive in that it is a purposeful selection of data sources with respect to their ability to provide information. Data are collected until saturation, i.e. until there is no more data to provide new elements (Sandelowski M. Sample size in qualitative research. Res Nurs Health. 1995;18(2):179–83).

The authors changed the conclusion (line 294-300).

This manuscript is a resubmission of an earlier submission. The following is a list of the peer review reports and author responses from that submission.

Round 1

Reviewer 1 Report

The authors present a study to investigate the impact of novel drug treatment for OSA on couples' lives. Ten patients on new drug therapy (reboxetine plus oxybutynin) for OSA were interviewed along with six of their partners. The results indicate that the drug treatment of OSA has a favourable impact on the couple's life. Although this study provides an avenue for other non-pharmacological treatments, there are several flaws noted.

1. What and how was OSA graded? What is the indication of the drug therapy?

2. How long was the drug therapy given prior to the interview?

3. How was the participant's comorbid addressed in this condition?

4. There are extensive grammatical errors found throughout the manuscript

Reviewer 2 Report

We have to congratulate to the authors for this manuscript based on subjective feelings of patients after using one week of medication to treat OSA. Due to these patients are enrolled in a RCT n the RebOx study we cannot  understand why these interviews have not been used in the control group. The scientific evidence of the subjetive improvement of these patients is relatively low, also due to the few patients entered in the study. We recommend authors to perform the same investigation comparing with the impressions obtained with the control group and evaluate if there are any statistical significances in the findings.